# Antimicrobial and Antivirulence Impacts of Phenolics on Salmonella Enterica Serovar Typhimurium

**DOI:** 10.3390/antibiotics9100668

**Published:** 2020-10-03

**Authors:** Zabdiel Alvarado-Martinez, Paulina Bravo, Nana-Frekua Kennedy, Mayur Krishna, Syed Hussain, Alana C. Young, Debabrata Biswas

**Affiliations:** 1Biological Sciences Program-Molecular and Cellular Biology, University of Maryland-College Park, College Park, MD 20742, USA; zalvara1@umd.edu; 2Department of Animal and Avian Sciences, University of Maryland, College Park, MD 20742, USA; pbravoro@terpmail.umd.edu (P.B.); frekua@terpmail.umd.edu (N.-F.K.); ayoung21@terpmail.umd.edu (A.C.Y.); 3Department of Public Health, University of Maryland, College Park, MD 20742, USA; mkrish98@umd.edu (M.K.); shussa98@terpmail.umd.edu (S.H.); 4Center for Food Safety and Security Systems, University of Maryland, College Park, MD 20742, USA

**Keywords:** *Salmonella enterica* serovar Typhimurium, phytochemicals, phenolic acids, antimicrobials, gallic acid, protocatechuic acid, vanillic acid

## Abstract

*Salmonella enterica* serovar Typhimurium (ST) remains a major infectious agent in the USA, with an increasing antibiotic resistance pattern, which requires the development of novel antimicrobials capable of controlling ST. Polyphenolic compounds found in plant extracts are strong candidates as alternative antimicrobials, particularly phenolic acids such as gallic acid (GA), protocatechuic acid (PA) and vanillic acid (VA). This study evaluates the effectiveness of these compounds in inhibiting ST growth while determining changes to the outer membrane through fluorescent dye uptake and scanning electron microscopy (SEM), in addition to measuring alterations to virulence genes with qRT-PCR. Results showed antimicrobial potential for all compounds, significantly inhibiting the detectable growth of ST. Fluorescent spectrophotometry and microscopy detected an increase in relative fluorescent intensity (RFI) and red-colored bacteria over time, suggesting membrane permeabilization. SEM revealed severe morphological defects at the polar ends of bacteria treated with GA and PA, while VA-treated bacteria were found to be mid-division. Relative gene expression showed significant downregulation in master regulator *hilA* and *invH* after GA and PA treatments, while fliC was upregulated in VA. Results suggest that GA, PA and VA have antimicrobial potential that warrants further research into their mechanism of action and the interactions that lead to ST death.

## 1. Introduction

Death because of enteric infections was found to be the third highest transmittable cause of death in the world, with diarrheal disease being the main source and infection by the bacterial pathogen *Salmonella* spp. responsible for at least 18.7% of those deaths [1]. Even though incidence is most pervasive in developing parts of the world, enteric disease caused by foodborne pathogens still accounts for substantial cases of illness, hospitalization, death and economic loss in the USA [2]. Of the major 31 pathogens associated to foodborne illness in the USA, *Salmonella enterica* is the leading bacterial etiological agent as it is estimated to be responsible for over 1 million cases of illness, 15,000 hospitalizations, 300 deaths, and an estimated loss of 3.5 billion dollars yearly, associated with the loss of productivity that accounts for cost of care, treatment, recovery and hours of work lost [3,4]. Though many of the already identified and sequenced *Salmonella enterica* serovars (>2500) have been linked to disease, *Salmonella enterica* serovar Typhimurium (ST) remains one of the most clinically relevant strains because of its ubiquitous presence in produce, meat and the environment, as well as being a major source of outbreaks and disease [5,6,7].

Though self-limiting for healthy individuals, at-risk populations such as children, elders and immunocompromised individuals can experience more invasive forms of the pathogen, requiring more intense treatment with antibiotics [8]. Invasive *Salmonella enterica* serovar Typhimurium (ST) infections are mediated through the activation of the genes found in the *Salmonella* Pathogenicity Island 1 (SPI-1), which code for an assembly of proteins known as the Type III Secretion System (T3SS) that aid in the attachment and subsequent invasion of host cells [9]. The intracellular survival of the bacteria is later achieved through the activation of the genes found in SPI-2, allowing it to avoid the immune system and resist lysis [10]. Strains found to be resistant to antibiotics have been corelated with high virulence and a more aggressive invasive capability towards their host, which subsequently results in a harder to treat infection [11,12,13]. The continuous and increasing discovery of antibiotic-resistant ST isolates from human samples [14], as well as in farm animals meant for consumption, poses a growing public health threat, greatly exacerbated by the need for novel antimicrobial compounds that do not add to the development of resistance.

New antimicrobial agents need to be accessible, easy to synthesize or extract, and effective, but must also be sustainable by not contributing to the rise of antibiotic resistance in ST, or in other bacteria by way of horizontal gene transfer and the antimicrobial resistome [14]. Plants and their byproducts are important candidate sources in the discovery of novel antimicrobials, primarily because of their availability, diversity of compounds and their complex chemical makeup [15,16]. Extracts prepared from various plant sources have been found to contain a vast array of bioactive organic small molecules with antimicrobial properties, particularly polyphenols. These polyphenols are often consumed as a part of ingesting foods that contain them, such as grains, vegetables, herbs and fruits. The range of species and concentration of polyphenolic compounds that can be found in these foods can vary among them and even within the same food group, as these are subject to external factors such as the growth conditions of the plant, environmental parameters, harvesting procedure, and extraction method [17]. Studies reporting on phenolic concentration within specific food groups have found cereals to contain a range from 26 to 3300 mg per 100 g (depending on the cereal), cocoa to range from 1204 to 4437 mg per 100 g, tea beverages to range from 29 to 103 mg per 100 mL, and fruits to range from 15 to 2556 mg per 100 g (depending on the fruit) [18,19].

Previous research on plant extracts with antimicrobial properties found that berry pomace, a byproduct containing the seeds and shell of the berry fruit after juicing, from blueberries (*Vaccinium corymbosum*) and blackberries (*Rubus fruticosus*) contains a high concentration and variety of polyphenols, which can be extracted economically [20]. This berry pomace extract (BPE) was shown to have significant antagonistic effects on ST growth and other cellular functions related to virulence [21]. The two major polyphenolic compounds that make up the majority of the phenolic content in plant extracts, which could be responsible for conferring antimicrobial effects, are flavonoids and phenolic acids. Though the exact mechanism of action is not yet understood, research on the antimicrobial capabilities of phenolic acids has shown potential, particularly against Gram-negative bacteria, since it is believed that they can diffuse passively through the outer membrane, bypassing one of the main defensive barriers against conventional antibiotics and making their way into the cytoplasm, which will lead to acidification and cell death [22,23,24]. Though membrane damage has been cited as one of the main outcomes of phenolic acid treatments, the events that lead to this damage and eventual cell death in ST treated with gallic acid (GA), protocatechuic acid (PA) and vanillic acid (VA), which are some of the most abundant phenolic acids in plants and in BPE, have not been well documented and the mechanism has not been elucidated [25,26].

This study evaluates the antimicrobial potential of three phenolic acids known to be in BPE, with the aim of comparing their individual effectiveness against ST. This will be determined by measuring patterns of growth inhibition, cell membrane permeability and visible morphological changes, as well as impairment of cellular functions related to virulence. These findings will serve as a base for discovering the mechanism of action of some of the components that make up BPE, allowing for better guided use of this product in the future, as well as understanding the antimicrobial potential of the phenolic acid family of compounds against increasingly resistant pathogens such as ST.

## 2. Results

### 2.1. Antimicrobial Effect on Bacterial Growth

A microdilution assay was used in order to determine the minimum inhibitory concentration (MIC) and minimum bactericidal concentration (MBC) of each individual compound and to determine phenolic acids with bactericidal/bacteriostatic antimicrobial potential when used individually against ST. The MIC was taken as the concentration at which there was no visible growth of bacteria in LB broth (Table 1). The MICs of GA, PA and VA were 3.5, 2.0 and 1.5 mg/mL, respectively. The MBC was determined as the concentration at which bacteria showed no detectable growth on LB agar after plating from LB broth aliquots. The MBCs of GA, PA and VA were 4.5, 2.0 and 2.0 mg/mL, respectively. (Figure 1). These concentrations exhibited the most acute antimicrobial effect; however, significant reduction as compared to the control was also recorded at lower concentrations. GA exhibited a statistically significant (*p* < 0.05) inhibition of bacteria up to 2.5 mg/mL; PA exhibited a statistically significant inhibition up to 1.5 mg/mL; VA exhibited a statistically significant inhibition up to 1.0 mg/mL. Calculating the ratio of MBC:MIC found the three compounds to be bactericidal, as they fell on or below a ratio of 2. 

### 2.2. Evaluation of Resistance Development

Bacteria were transferred from a well containing sub-MICs of each compound that permitted substantial bacterial growth over time. Transfers were performed 12 times (generations) while the MIC from the aliquot was recorded and used as the indicator for changes in susceptibility or resistance (Figure 2). Though no continuous or consistent increase during this period was shown, by the end of the 12 passages, the MIC values for GA stabilized at 3.5 mg/mL, while PA increased to 2.5 mg/mL, showing an increase of 1 mg/mL from the beginning of the experiment, while VA remained at 1 mg/mL from the 4th passage.

### 2.3. Plasma Membrane Permeability and Damage

Changes in outer membrane permeability were measured using BacLight^TM^ Bacterial Viability Assay, which uses SYTO9 and propidium iodide (PI) as indicators of membrane integrity. SYTO9 freely crosses the cell membrane and intercalates with DNA, emitting a green fluorescence, while PI is used as the main indicator for membrane damage, since it mainly crosses the bacterial membrane when there is increased permeabilization, allowing it to compete with SYTO9 for DNA intercalation within the bacterial cytoplasm, leading to the emission of red fluorescence that can be measured spectrophotometrically in units of relative fluorescent intensity (RFI) and microscopically (Figure 3 and Figure 4). There was no significant difference between control and treatments at 0 h. Significant (*p* < 0.05) differences between the RFI levels of the control and treatment were seen in GA at subsequent timepoints, with an increase of 2584, 2123 and 2345 RFI at 4, 8 and 24 h. Overall RFI for GA decreased at 8 and 24 h, but remained significantly higher as compared to the control. The RFI for PA and VA experienced an increased to 1851 and 4062 RFI, respectively, at 4 h, which was significant (*p* < 0.05) as compared to the control. At 8 h, the RFI for both PA and VA decreased, with only the latter remaining slightly numerically higher than the control. At 24 h, both RFI values increased to 2441 and 3212 for PA and VA, respectively, as well as remaining significantly higher than the control. The results from fluorescent microscopy (Figure 4) showed an untreated bacterium to mostly take up SYTO9, as evidenced by the prevalent green coloration of the bacteria, though the presence of slightly more yellow bacteria at increasing timepoints signals a low take-up of PI. In treated bacteria, an increasing PI buildup could be seen at 4 h particularly for GA and VA, while at 8 h all bacteria exhibited a PI buildup and higher counts of red bacteria. At 24 h, all treated bacteria had comparatively more presence of red bacteria than green.

### 2.4. Morphological Alterations of ST

Physical changes in ST were examined using scanning electron microscopy (SEM), which allows for a detailed visualization of the bacterial outer membrane after treatment and a later comparison to the untreated control (Figure 5). The untreated control exhibited the usual rod shape associated with ST. However, a distinct inward collapse of the cell membrane, that could be visualized as a hole, was observed at the polar ends of rods treated with GA and PA. On the other hand, VA did not exhibit the same morphological defect as its other two phenolic counterparts; however, cells from this sample featured a vast number of rods in the middle of binary fission, with a clearly formed division ring.

### 2.5. Relative Gene Expression of ST

Changes in the expression of ST virulence genes were evaluated through the use of qRT-PCR and gene-specific primers for *fliC*, *hilA, hilD, invH*, *prgH, prgK* and *sipA*. When compared to an untreated control, samples from bacteria treated with individual phenolic acids mostly showed a downregulation of examined virulence genes (Figure 6). All virulence genes from bacteria treated with GA showed downregulation when compared to the untreated control, with *hilA, hilD, invH, prgH, prgK* and *sipA* being significant (*p* < 0.05) by 2.51, 3.02, 2.35, 2.24, 2.33 and 2.06 log-folds, respectively. PA-treated samples demonstrated statistically significant (*p* < 0.05) downregulation in *fliC*, *hilA* and *invH* by 1.87, 2.0 and 0.78 log-folds, respectively, with only a numerical decrease in all but *prgH*. Samples treated with VA showed significant upregulation for *fliC* with an increase of 2.35 log-folds, while the rest of the genes showed a numerical decrease.

## 3. Discussion

The use of plant-derived polyphenolic compounds against Gram-negative pathogenic bacteria has gained notice because of the need for novel antimicrobial agents that can counteract the increasing trends of antibiotic resistance. Though polyphenolic compounds belonging to groups such as flavonoids, tannins and catechins have received much attention for their antimicrobial potential, many researchers have focused on compounds belonging to phenolic acids. These comprise at least one-third of the polyphenolic compounds found in plants, are readily solubilized without need for toxic solvents, and have been proven to have antibacterial potential [27,28]. The low molecular weight of phenolic acids compared to other polyphenols has been cited as a factor that potentially allows for more interactions with bacterial structures, resulting in greater antimicrobial effects [29]. Though the compounds evaluated in this study share a strong structural similarity, differences in their functional groups could play an important role when it comes to efficiency, half-life and target specificity [30]. Hydroxyl (-OH) and methoxy (-OCH_3_) groups have been cited as key attributes that affect molecular interactions, as these create differences in polarity and oxidation rates [31].

The phenolic acids evaluated in this study have been previously noted by other researchers, primarily for their roles as antioxidants, documented use as food preservatives, and antimicrobial potential against Gram-negative bacterial pathogens [32]. GA, PA and VA specifically have been shown to inhibit the growth of *Pseudomonas* spp., *Listeria monocytogenes*, *Mannheimia haemolytica*, *Pasteurella multocida*, *Escherichia coli* and some strains of *Salmonella* spp., as well as being able to reduce the pathogenicity of other bacteria such as *Proteus mirabilis* [33,34,35,36]. However, the effectiveness against ST, specifically, has not been consistently documented, in addition to the mechanisms of action underlying the molecular interactions that precede cell death remaining poorly understood. Previous research that has delved more profoundly into studying the effects that phenolic acids have on Gram-negative bacteria have cited a decreased metabolic activity, inhibition of enzyme function and cell membrane damage as common outcomes of exposure to these compounds [37]. Though these could all contribute to cell death, outer membrane damage has been consistently documented, which warrants further study, since the outer membrane of Gram-negative bacteria is of extreme importance for survival in adverse environments and resistance to antibiotics [38].

The results from this study suggest that GA, PA and VA have concentration-dependent antimicrobial capability against ST, with all of them being able to inhibit bacterial growth to the point of non-detection at specific concentrations, as exhibited by the MBC values (4.5, 2 and 2 mg/mL, respectively). GA was outperformed by PA and VA in both MIC and MBC tests, with PA and VA requiring lower concentrations of the compound to significantly inhibit growth. However, it is important to note that bacteria treated with VA were further challenged by the ethanol used for preparing the solution. Considering the factors mentioned above, PA could be considered the better of the three compounds as it is more potent and exhibits its effects without interference from additional solvents.

The emergence of antimicrobial resistance poses one of the main concerns and challenges for novel antimicrobial discovery, development and sustainability [39]. The use of individual compounds in particular has become increasingly challenging as research has demonstrated that bacteria have intrinsic mechanisms that serve to minimize susceptibility to a variety of known bioactive molecules, in addition to having the ability to adapt to new ones [40,41]. In this study, resistance to individual compounds was measured throughout 12 passages and found ST to still be susceptible to all phenolic compounds at the end of the passages. Even though there was an increase of 1 mg/mL for the MIC value of GA and PA, the concentration did not increase beyond that point. VA’s MIC remained stable from early on in the experiment and showed no significant increase at the final passage. The development of resistance could be related to how easily the compound could be oxidized by environmental factors or by the bacteria [42]. The methoxy group of VA makes it more stable and less prone to oxidation than GA and PA, which are more susceptible to direct oxidation of their respective hydroxyl groups by bacteria and to the effects of media acidification as a result of bacterial growth [43]. With this in mind, the observed increase in resistance to GA and PA could be an indirect result of changes in the metabolic pattern of bacteria and not a direct response to the compounds [32]. Future studies will involve more extensive and numerous passages, as well as experiments that further elucidate the mechanism of action and bacterial response, which will be important for preventing the risk of developing resistance and potentially novel ways in which it can be actively counteracted.

The outer membrane of Gram-negative bacteria serves as the first defensive barrier against antimicrobials and is responsible for conferring resistance to a variety of antibiotics, as well as to multiple environmental stressors [44]. Research on food contaminated with *E. coli* has discovered that the communities found in these environments resemble stationary phase bacteria that are well adapted to a lower pH, higher temperatures and osmotic pressure [45]. This potentially means that more virulent pathogens can survive the passage through the gastrointestinal tract and are also initially more virulent. However, one of the characteristics of these bacteria is an increased re-enforcement of the cell envelope [46]. On the other hand, resistance to cationic antibiotics that target outer membranes has been reported to be usually mediated through the modification of lipopolysaccharides (LPS), leading to an alteration of the net charge of the outer membrane [47]. Phenolic acids have long been hypothesized to damage bacterial outer membranes through mechanisms mediated by electrostatic and ionic interactions, or by passive diffusion to the cell interior, but the exact interaction that leads to this outcome has remained undefined [25]. Previous studies have reported this effect to be dependent on the relative acidity and oxidation of the compound in the solution at the time of contact with the bacteria, which will affect whether it will be able to penetrate the outer membrane and the extent to which it will interact with bacterial lipids and proteins located in the cytoplasm and periplasmic space [25,26]. Measuring PI RFI served as an indicator to measure changes in membrane permeability as a result of treatment and the time of exposure. This experiment demonstrated a significant increase for all treatments at the 4 and 24 h timepoints when compared to the untreated control at that same time, with GA and VA accounting for the highest reported RFI reads, though at 24 h PA had similar value as GA. The reasons behind the decrease in RFI at 8 h is unclear, but could be related to a delay in growth and cell death because of the treatments, yielding a lower cell count that resulted in a lower RFI signal for PI, but leaves open the possibility that this occurred as a result of some acid tolerance response mechanism. Previous research on membrane-permeable organic acids suggests that small molecules such as phenolic acids readily cross the outer membrane and damage the cell from the inside through cytoplasm acidification, which is known to be a factor in increasing membrane permeability [48]. Another review on weak acid interactions with bacteria suggested that when weak acids disassociate, there is an accumulation of anions in the periplasmic space that can lead to a disruption of membrane function and metabolic processes, despite the various resistance mechanisms that ST has developed to counteract weak acids [49]. The time that this mechanism takes to activate and take effect is unclear, but a decrease in PI uptake at 8 h could be a result of the activation of these tolerance mechanisms, but an increased exposure time overwhelms the bacteria leading to a resurgence of PI uptake at later timepoints such as the one seen at 24 h.

Further investigation on alterations to membrane integrity were examined through microscopy. Through fluorescent microscopy, there was a visible representation of the increasing membrane permeabilization as the exposure time to the treatments increased, evidenced by the increasingly red coloration of treated cells as time passed. Further investigation on this effect was conducted through SEM, which allows for a detailed viewing of cell morphology. Images taken of GA and PA treatment showed bacteria that shared a similar morphology, suggesting a similar mode of action of the compounds and a similar cellular response, even though the lethal dose required for achieving cell death was significantly higher for GA, as explained previously. The most notable cellular defects that were observed in bacteria treated with GA and PA were inward dents located at the polar ends of the rod-shaped cell, while the central body of the cell remained relatively unchanged as compared to an untreated control. The reason for this change is yet to be determined, as this kind of phenotype has not been reported previously for bacteria treated with phenolic acids. Possible explanations for this particular morphology could be related to damage or alterations to proteins more commonly collocated at cellular poles, such as osmotic pressure regulators and division complexes [50], or could be related to changes in the integrity of bacterial cytoskeletal proteins [51,52]. The morphotype observed for VA was different than the other phenolic acids, as these did not show clear evidence of membrane damage, or cell structure change; however, the overwhelming majority of bacteria observed were found to be mid-division. These contained a clearly formed division septum, but separation of the cell membrane was arrested as the daughter cell remained attached. These findings suggest a perturbation of the cell division process that allows for septum formation but prevents finalization.

In addition to growth inhibition and cell death, a reduction in virulence as a result of treatment is a desired outcome, especially for ST cases, which are known to be invasive pathogens and use these strategies to prolong survival in the host and create further health complications [53]. Relative expression was measured for genes responsible for synthesis and the assembly of the T3SS, specifically *hilA*, *hilD*, *invH*, *prgH* and *prgK* as well as genes related to other aspects of virulence such as *fliC*, which increases motility and *sipA* which is responsible for intracellular survival. The results of this experiment demonstrated a downregulation after individual treatment with GA, PA and VA in most of the genes that were tested for, with the exception of the notable upregulation of *fliC* in VA and a slight numerical increase in *prgH* in PA. The most significant reduction in all genes was in bacteria treated with GA as all genes related to the structural assembly of the T3SS that were tested for were downregulated, with the addition of one related to intracellular survival. Downregulation in PA was more evident for *fliC*, which is important for motility, but also in *hilA* and *invH*, the former being a key master regulator for the initiation of the T3SS and the latter being a pilot protein responsible for the co-localization of the base of the needle in the bacterial inner membrane. Though the proposed mechanism of action for these compounds is damage to the membrane and bacterial structure, previous research has reported that damage to the envelope of ST, particularly alterations to the LPS of the outer membrane, leads to the downregulation of genes related to flagellum synthesis and assembly [54]. The same study found other virulence-related genes from SPI-1 and SPI-2 to be downregulated. To our knowledge, the specific effects of phenolic acids on changes in virulence have not been examined before for ST, leaving room for further exploration by applying these products for reducing the incidence of invasive salmonellosis, in addition to revealing one of the potential routes by which plant extracts reduce virulence and could have protective qualities towards host cells, as has been reported in the past [21,55]. In this specific experiment, GA demonstrated the greatest amount of promise as it was able to reduce the expression of most virulence genes, though PA should not be discounted since it also reduced key regulatory genes. The way in which VA impacts virulence is yet unclear, though an increase in *fliC* could be a sign of stress that could warrant deeper investigation into that possible mechanism. These differences in outcome are favorable if in the future there is an attempt to use these compounds synergistically.

Plant extracts are promising candidates because the diversity of bioactive compounds they contain are thought to exert multiple adverse actions over a given bacterium, reducing the risk of developing resistance [56]. However, understanding the mechanisms of action of the individual constituents of these extracts will allow for their use in a more guided and efficient manner, even allowing for the possibility of rescuing conventional antibiotics [57,58]. The antimicrobial efficiency of individual phenolic compounds is not as effective as that of leading antibiotics or to that of many raw plant extracts, but in the future may provide alternatives to infections considered as untreatable by antibiotics. This study also presents a way to uncover new targets and mechanisms that could be further exploited when developing and evaluating novel antimicrobials, in addition to exploring the cellular response to weak phenolic acids that lead to the changes in shape and gene expression patterns that were seen in this study, which have not been reported in the past.

## 4. Materials and Methods

### 4.1. Bacterial Strain and Their Growth Conditions

In this study, *Salmonella enterica* serovar Typhimurium (ATCC 14028) (ST) was used. ST was grown in Luria-Bertani (LB) (Becton, Dickinson and Co., Franklin Lakes, NJ, USA) agar at 37 °C under aerobic conditions (Thermo Fisher Scientific Inc., Marietta, OH, USA). All experiments described were performed in LB broth (Becton, Dickinson and Co., Franklin Lakes, NJ, USA) and were incubated under aerobic conditions for 24 h in 37 °C with continuous aeration at 150 rpm through the use of a shaking incubator.

### 4.2. Compounds and Stock Solution Preparation

GA (Acros Organics, Geel, Belgium), PA (Sigma-Aldrich, St. Louis, MO, USA) and VA (Alfa Aesar, Ward Hill, MA, USA) were purchased in solid powder form. Stock solutions of gallic acid and protocatechuic acid were prepared by dissolving in sterile deionized water, while vanillic acid was prepared by dissolving in 30% ethanol (Pharmco-Aaper, Brookfield, CT, USA). Stocks for all compounds were prepared to a concentration of 10 mg/mL. Phosphate buffer saline (PBS) was prepared to a pH of 7.2.

### 4.3. Determining MIC and MBC

The MIC and MBC values for individual phenolic compounds were determined using the broth microdilution method as described by Clinical and Laboratory Standards Institute Performance Standards for Antimicrobial Susceptibility Testing M100 [59] and performed previously [21], with slight modifications. Briefly, an overnight culture of ST cultured on LB agar was used to prepare a bacterial suspension in PBS fixed to an optical density (OD_600_) of 0.1 (8 log CFU/mL). This bacterial suspension was further diluted and used to inoculate 24-well plates containing LB broth with increasing concentrations of phenolic acids (from 0.5 to 4.5 mg/mL) to achieve a final bacterial load of 4 log CFU/mL in each well. Controls were prepared to have an equivalent media-to-solvent ratio with their respective treatment counterpart. MIC was determined as the lowest concentration at which there was no visible bacterial growth in the well after 24 h. MBC was determined as being the lowest concentration after 24 h at which there was ≥99.9% eradication of bacteria after plating in LB agar. The MBC:MIC ratio was calculated in order to determine if the treatments were bactericidal (<2) or bacteriostatic (>16).

### 4.4. Resistance Development over Time

Development of resistance to the phenolic acids in ST was evaluated using methods previously described on resistance development to antibiotics with modifications [60]. Briefly, a 24-well plate was prepared following the same set up as the microdilution assay. The bacteria from the well with the highest concentration of each of the respective treatments, below MIC, and that allowed growth after 24 h, were selected. An aliquot from the bacteria in this well was taken and again fixed to an OD_600_ of 0.1 to repeat the process of inoculation of a new 24-well plate. This was performed over 12 generations and changes in the MIC were recorded over this time.

### 4.5. Measuring Membrane Damage and Permeability Using Fluorescent Dyes

Damage to the cell membrane of ST that might have led to increased permeability of the membrane during treatment with phenolic acids was evaluated through the use of a Cytation 3 fluorescence microplate reader (BioTek, Winooski, VT, USA) to measure RFI and later fluorescent microscopy was performed using the Zeiss AxioObserver Florescence microscope (Carl Zeiss AG, Jena, Germany). Nucleic acid dyes SYTO9 and PI from a BacLight^TM^ Bacterial Viability Kit (Invitrogen Molecular Probes, Carlsbad, CA, USA) were used as indicated by the manufacturer, with some modifications. Briefly, ST cultures set to a final OD_600_ of 0.1 in LB broth combined with lethal doses from the previously calculated MBC doses of each compound, along with a negative untreated control, were individually treated for 0, 4, 8 and 24 h, at which times 1 mL aliquots were taken from the total suspension. At the given timepoints, aliquots were prepared as per the manufacturer recommendations by spinning down the samples, washing with and resuspending in a 0.85% NaCl solution. SYTO9 and PI were both added to the samples and incubated for 15 min, after which the fluorescence of both dyes was measured in a black 96-well plate with a fluorescent plate reader set to an excitation of 485 nm with emissions of 530 mm and 630 nm for SYTO9 and PI, respectively. In addition to the plate reading, 5 μL aliquots from each sample at 4, 8 and 24 h timepoints were taken and fixed using 10% formaldehyde for preparing slides that were later observed using fluorescent microscopy. Pictures taken with the Zeiss AxioObserver Florescence microscope were minimally processed using the default Zeiss ZEN software from the same company.

### 4.6. Scanning Electron Microscopy

Preparation of samples for observing changes in the cell shape of ST through SEM was performed as described previously [61,62], with some modifications. Briefly, ST cultures in LB broth were individually treated for 24 h with sublethal concentrations (below calculated MIC, namely 3.0, 1.5 and 1.0 mg/mL for GA, PA and VA, respectively) of phenolic compounds, in addition to an untreated control. Bacteria were pelleted, washed and resuspended in PBS, after which samples were placed in a polycarbonate membrane filter (GTTP 0.2 μm) (MilliporeSigma, Rockville, MD, USA) and fixed with 2.5% (*v/v*) glutaraldehyde (Electron Microscopy Sciences, Hatfield, PA, USA) for 1 h. Filters were washed with deionized (DI) water and later dehydrated by sequentially soaking with increasing concentrations of absolute ethanol (10%, 20%, 50%, 80% and 100%) for 5 min each. Filters were kept overnight over anhydrous magnesium sulfate to eliminate residue moisture. Filters were prepared for SEM by sputter-coating with gold and observed using the Hitachi SU-70 FEG Scanning Electron Microscope (Hitachi High-Tech Corporation., Minato-ku, Tokyo, Japan).

### 4.7. RNA Extraction and cDNA Synthesis

Samples were prepared for RNA extraction by inoculating a bacterial suspension of ST to a final concentration of 4 log CFU/mL and incubated overnight. They were individually treated with the sublethal concentrations (below calculated MIC, namely 3.0, 1.5 and 1.0 mg/mL for GA, PA and VA, respectively) of each compound (concentration before MIC) and were subsequently collected. RNA was extracted following the previously described methods with modifications [63]. Briefly, samples were centrifuged at 14,000× *g* for 15 min to collect bacterial pellet, and TRI Reagent^®^ (Molecular Research Center Inc., Cincinnati, OH, USA) protocol was used to homogenize pellet and isolate the RNA sample. RNA concentration in the sample was measured using a NanoDrop spectrophotometer (Thermo Fisher Scientific Inc., Marietta, OH, USA) for standardization and to use as the template for cDNA synthesis with the High-Capacity cDNA Reverse Transcription Kit (Applied Biosystems, Foster City, CA, USA), in accordance with the manufacturer indications. The cDNA synthesis protocol was set to incubate for 25 °C for 10 min, 37 °C for 120 min, and 85 °C for 5 min.

### 4.8. Quantitative RT-PCR Assay

Changes in the relative expression of genes related to the virulence of ST were measured using qPCR. The qPCR reaction was prepared in accordance with the manufacturer indications of the PerfeCTa SYBR Green Fast Mix protocol (Quanta Bio, Beverly, MA, USA), and performed using the Eco Real-Time PCR system (Illumina, San Diego, CA, USA). Cycle protocol was set to 30 s at 95 °C, followed by 40 cycles of 5 s at 95 °C, 15 s at 55 °C and 10 s at 72 °C. Custom primer sequences corresponding to the conserved regions of respective virulence genes in ST were used to compared to a reference house-keeping gene belonging to 16S-rRNA (Table 2). 

### 4.9. Statistical Analysis

Data were analyzed using a Student’s t-test to determine the significant difference (*p* < 0.05) between the untreated control and the separate treatment with the individual compound.

## 5. Conclusions

In the current study, widely distributed phenolic acids—GA, PA and VA—were tested for their antimicrobial potential against ST. Results show that these compounds have the capacity to inhibit the growth of ST and act as a bactericidal in a concentration-dependent manner. After 12 passages, there was no significant development of resistance in ST to these compounds. The increased uptake in PI measured in the fluorescent plate reader and observed through fluorescent microscopy suggests that the membrane is permeabilized as a result of exposure to phenolic compounds over increasing bouts of time. Alterations to the cell membrane were confirmed with SEM, in which there were structural deformations in treated cells. GA and PA lead to the formation of dents at the polar ends of the bacteria, whereas VA caused an arrest in the cleavage between dividing cells. These morphological changes provide insight into the mechanism of action of these compounds and the cellular responses that they induce. In addition to this, GA and PA led to the significant downregulation of important virulence gene regulators—though VA only showed a numerical decrease, it might have other effects over ST. This knowledge can be used in the future to develop novel therapeutic approaches and techniques that exploit the effects that phenolic acids have over ST, in order to control the pathogen and reduce the severity of infections.

## Figures and Tables

**Figure 1 antibiotics-09-00668-f001:**
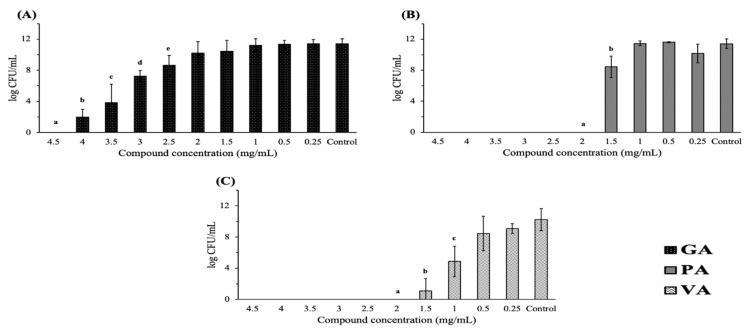
Concentration-dependent antimicrobial evaluation for individual phenolic compounds by determining minimum bactericidal concentration (MBC) (log CFU/mL) for *Salmonella enterica* serovar Typhimurium (ST) treated with increasing concentrations (0.25–4.5 mg/mL) of gallic acid (GA) (**A**), protocatechuic acid (PA) (**B**) and vanillic acid (VA) (**C**). Letters denote statistically significant difference (*p* < 0.05) as compared to control and between concentrations (a–e).

**Figure 2 antibiotics-09-00668-f002:**
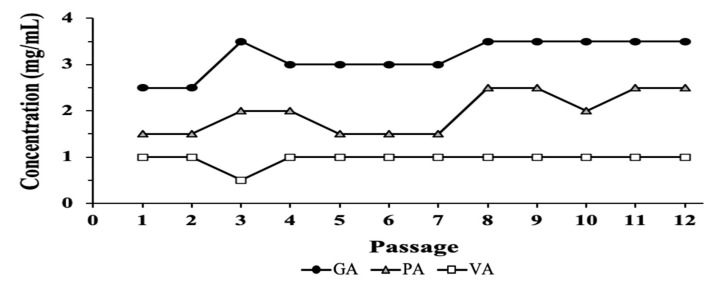
Evaluation of changes in susceptibility of ST individually treated with GA, PA and VA over 12 generations.

**Figure 3 antibiotics-09-00668-f003:**
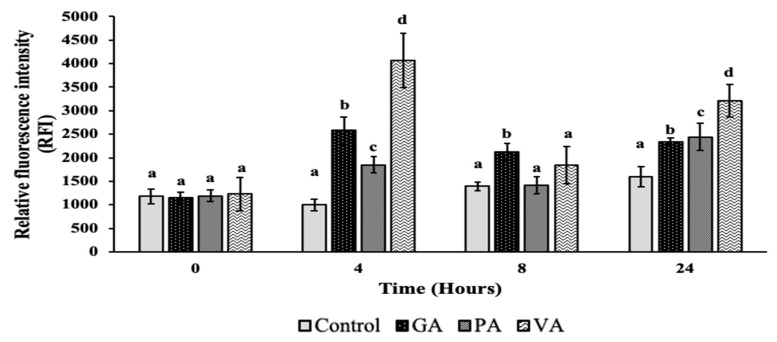
Measuring permeability of cells using fluorescence microplate reader measurements of relative fluorescent intensity (RFI) for untreated and treated ST with GA, PA and VA. Letters (**a**–**d**) denote significant difference between treatment groups (*p* < 0.05).

**Figure 4 antibiotics-09-00668-f004:**
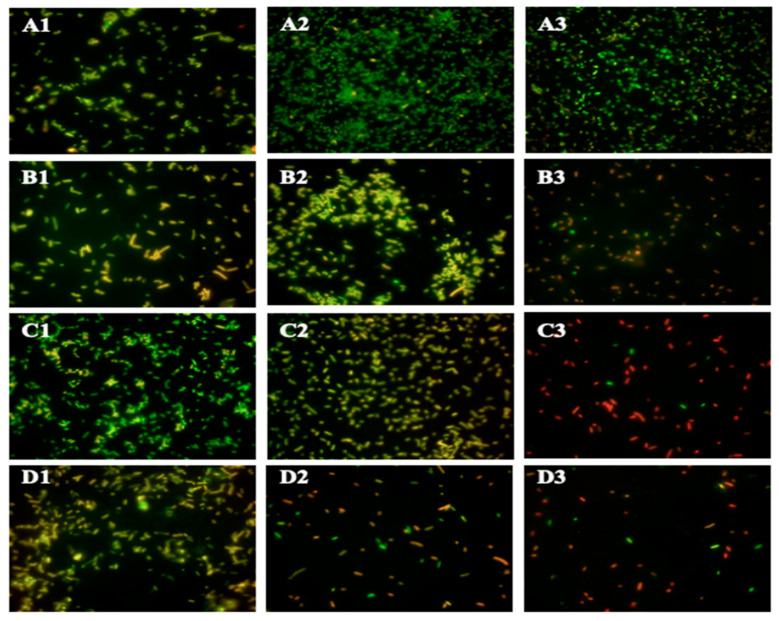
Observing changes in membrane permeabilization through fluorescent microscopy with dyes SYTO9 and PI for negative control (**A1–A3**), GA (**B1–B3**), PA (**C1–C3**), VA (**D1–D3**) over 4 (1), 8 (2) and 24 (3) h of treatment periods.

**Figure 5 antibiotics-09-00668-f005:**
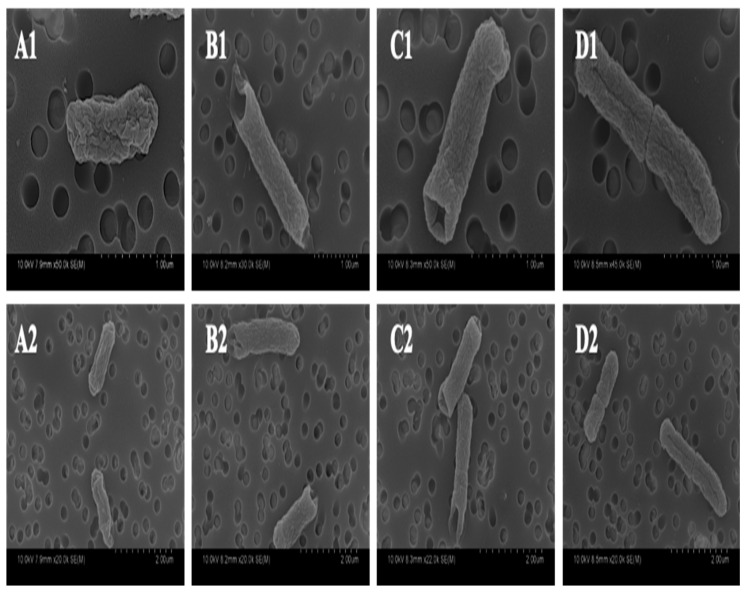
Scanning electron microscopy (SEM) images of untreated ST as negative control (**A1**,**A2**), as well as treated ST with GA (**B1**,**B2**), PA (**C1**,**C2**), VA (**D1**,**D2**), taken at 1 μm (**A1–D1**) and 2 μm (**A2–D2**) magnification.

**Figure 6 antibiotics-09-00668-f006:**
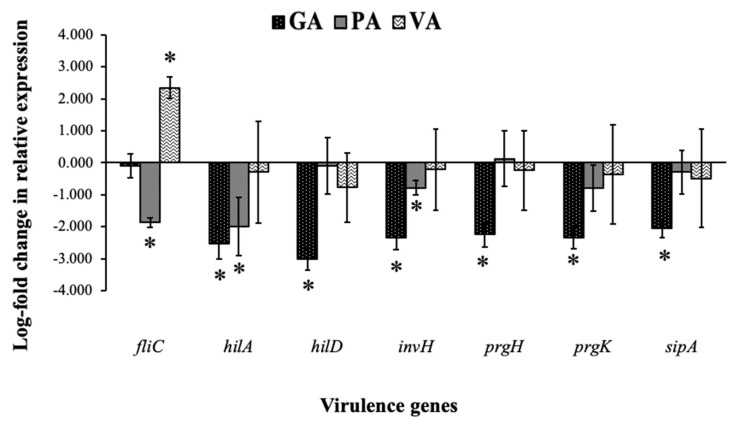
Evaluation of log-fold changes of ST virulence genes through measurements of relative gene expression. Significant difference in treatment as compared to control (*p* < 0.05) was denoted with an asterisk (*).

**Table 1 antibiotics-09-00668-t001:** Antibacterial effect of phenolic acids.

Treatment	MIC (mg/mL)	MBC/MIC	Bactericidal/Bacteriostatic
Gallic acid	3.5	1.28	Bactericidal
Protocatechuic acid	2	1	Bactericidal
Vanillic acid	1.5	1.33	Bactericidal

**Table 2 antibiotics-09-00668-t002:** Primers used in determining gene expression of ST strains.

Gene	Protein	Primer Sequence (5’-3’)
16S rRNA	16S ribosomal protein	F: GTAGTACGATGGCGAAACTGC R: CTTCTCGACCCGAGGGACTT
*fliC*	flagellum subunit	F: GCAGATGACGGTACATCCAA R: CCAGATCAGGCTGTGCTTTA
*hilA*	SPI-1 transcriptional regulator	F: AATGGTCACAGGCTGAGGTG R: ACATCGTCGCGACTTGTGAA
*hilD*	SPI-1 transcriptional regulator	F: CTCTGTGGGTACCGCCATTT R: TGCTTTCGGAGCGGTAAACT
*invH*	adherence and invasion	F: GGTGCCCCTCCCTTCCT R: TGCGTTGGCCAGTTGCT
*prgH*	T3SS needle support at membrane	F: TGAACGGCTGTGAGTTTCCA R: GCGCATCACTCTGACCTACCA
*prgK*	T3SS needle support at membrane	F: GGGTGGAAATAGCGCAGATG R: TCAGCTCGCGGAGACGATA
*sipA*	actin binding protein for cell invasion	F: CGTCTTCGCCTCAGGAGAAT R: TGCCGGGCTCTTTCGTT

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
