# Peer review of "Antimicrobial and Antivirulence Impacts of Phenolics on Salmonella Enterica Serovar Typhimurium"

_antibiotics, 2020, doi:10.3390/antibiotics9100668_

Round 1

Reviewer 1 Report

The bactericidal effects of the polyphenols gallic acid, protocatechuic acid and vanillic acid against Samonella were determined to be in the range of 1.5 to 4.5 mg/ml. Resistance development over 12 generations was very low. Permeability changes of the cellular membranes were detected with propidium iodide. Cell poles of cells treated with gallic acid or protocatechuic acid were observed to be collapsed when looked at in scanning electron microscopy. Vanillic acid treated cells showed often a division ring and intact cell poles.

The expression of genes encoding components of the T3SS system were found to be lowered in comparison to the untreated cells. Only the expression of the flagellin gene fliC was enhanced by vanillic acid.

The manuscript is interesting, however, I would like to see some details specified below.

Growth conditions in the 24 well plate – how was the aeration – more anaerobic? Growth curves with different concentrations of the polyphenols added during logarithmic growth could give a hint if some cells lyse when the cell pole collapses. Also the growth conditions used to study gene expression could be shown with an indication when the samples were drawn.

Line 306: I would like to see some examples – how much polyphenol is normally found in certain nutrients?

Line 358: when treated with the lethal dose of a polyphenol –did the OD change after 4, 8 and 24 h –what is meant with an aliquot – the same OD as at time point 0? or volume?

From the microscopic pictures one gets the impression that VA had a strong growth inhibiting effect visible in the 8 h sample while GA and CA showed lower cell numbers in the 24 h samples.

Line 374:  what was the sublethal concentration?

Line 386: what were the sublethal concentrations? In which growth phase the samples were drawn – most likely in stationary phase? How meaningful are those results? I expect differences between a sample from a logarithmically growing culture and a stationary phase culture.

Minor comments

Antimicrobial activity of vanillic acid has been described, this could be mentioned: Torzewska A, Rozalski A. in Microbiol Res. (2014)169:7-8

Inhibition of crystallization caused by Proteus mirabilis during the development of infectious urolithiasis by various phenolic substances.

Line 156:...as its other two ...

Line 264: ... that were observed ...

Line 307 ... over a given bacterium....

Line 372: “....morphological changes in the cell membrane ...” possibly better .... in the cell shape ...

Line 375: .... Bacteria were pelleted

Author Response

Responses to Reviewer 1 comments (Manuscript ID: antibiotics-945777)

Comments and Suggestions for Authors

The bactericidal effects of the polyphenols gallic acid, protocatechuic acid and vanillic acid against Samonella were determined to be in the range of 1.5 to 4.5 mg/ml. Resistance development over 12 generations was very low. Permeability changes of the cellular membranes were detected with propidium iodide. Cell poles of cells treated with gallic acid or protocatechuic acid were observed to be collapsed when looked at in scanning electron microscopy. Vanillic acid treated cells showed often a division ring and intact cell poles.

The expression of genes encoding components of the T3SS system were found to be lowered in comparison to the untreated cells. Only the expression of the flagellin gene fliC was enhanced by vanillic acid.

The manuscript is interesting, however, I would like to see some details specified below.

Growth conditions in the 24 well plate – how was the aeration – more anaerobic? Growth curves with different concentrations of the polyphenols added during logarithmic growth could give a hint if some cells lyse when the cell pole collapses. Also the growth conditions used to study gene expression could be shown with an indication when the samples were drawn.

Response: We appreciate reviewer’s positive words and comments for our manuscript. In response to this comment, we have clarified M&M section as “The 24-well plate was placed on a shaking incubator set to 37°C and 150 rpm, giving the bacteria continual aeration under aerobic conditions. These same growth conditions were also used for the rest of the experiments” in Line 345 of the revised manuscript.

Line 306: I would like to see some examples – how much polyphenol is normally found in certain nutrients?

Response: We appreciate the reviewer’s interest with regards to this topic. With this comment in mind, we have given further examples and information about this subject by expanding on it in the background section (Line 68) and have added the additional appropriate references (Line 514, 517 and 519).

Line 358: when treated with the lethal dose of a polyphenol –did the OD change after 4, 8 and 24 h –what is meant with an aliquot – the same OD as at time point 0? or volume?

Response: We appreciate the reviewer pointing out these questions. The OD for these samples was not measured again at later timepoints, as the focus was placed on preparing for fluorescent spectrophotometry and microscopy. On the other hand, an aliquot refers to a 1 mL portion that is taken form the total suspension of bacteria, media and treatment (Line 387), so the total volume is reduced with every timepoint.

From the microscopic pictures one gets the impression that VA had a strong growth inhibiting effect visible in the 8 h sample while GA and CA showed lower cell numbers in the 24 h samples.

Line 374:  what was the sublethal concentration?

Response: We appreciate this question and the reviewer’s willingness to clarity. In response to this comment, we have added the concentrations used for this and other experiment for which we use sub-lethal concentrations of the compounds (Line 401 and 413). The sublethal concentration of the compounds were used in order to allow bacteria to grow but remain under pressure from the compounds.

Line 386: what were the sublethal concentrations? In which growth phase the samples were drawn – most likely in stationary phase? How meaningful are those results? I expect differences between a sample from a logarithmically growing culture and a stationary phase culture.

Response: We are very thankful for the reviewer’s critical analysis of our experiments. These samples were indeed drawn at stationary phase for SEM and RNA extraction. It is possible that cellular response will be different at different growth stages. However, we wanted to evaluate the effects of the treatments at the stationary phase, since it has been reported that at this stage bacteria are more tolerant to external stressors, partly mediated through re-enforcing cell envelope structure and acid tolerance response. We also believed this to be relevant since it has been reported that the bacteria found in contaminated food mostly resemble stationary phase bacteria, which makes them more likely to survive the passage through the gastrointestinal tract and makes them potentially more virulent. This justification was added to the discussion (Line 247) and additional citation for this was provided (Line 594, 597 and 599).

Minor comments

Antimicrobial activity of vanillic acid has been described, this could be mentioned: Torzewska A, Rozalski A. in Microbiol Res. (2014)169:7-8

Inhibition of crystallization caused by Proteus mirabilis during the development of infectious urolithiasis by various phenolic substances.

Response: We value the reviewer’s suggestion and believe it adds valuable information to our manuscript. Accordingly, we have added this in the Discussion section and modified the references accordingly (Line 203 and Line 569).

Line 156:...as its other two ...

Response: We appreciate the reviewer for carefully pointing out this grammatical error. We have corrected “it” to say “its” (Line 166).

Line 264: ... that were observed ...

Response: We like to thank to reviewer for carefully pointing out this grammatical error. The original word “was” has been changed to the correct “were” (Line 287).

Line 307 ... over a given bacterium....

Response: We appreciate the reviewer pointing out this grammatical error. The original word “bacteria” has been changed to “bacterium” (Line 330).

Line 372: “....morphological changes in the cell membrane ...” possibly better .... in the cell shape ...

Response: We thank reviewer for this valued suggestion, which is why we have changed the phrase to that suggested by the reviewer (Lie 398).

Line 375: .... Bacteria were pelleted

Response: We are grateful to the reviewer for pointing out this grammatical error. It has been fixed in text of the manuscript as “Bacteria” (Line 402).

Thank you

Best regards

Reviewer 2 Report

The content and the results of the submitted manuscript are interesting, new and contribute to a wide knowledge on the use of polyphenolic compounds as antimicrobial agents against the pathogen Samonella enterica serovar Typhymurium so giving an insight for novel therapeutic systems as promising alternatives to the conventional antibiotics. The paper is well presented and the methods used adequate for describing the experiments. However some points are not clear.

Results

Line 107: “ ……as they fell on or below a ratio of 2”.

Line 136-137: “At 8 hr, RFI for both PA and VA decreased, only remaining slightly numerically higher than control”? From Figure 3 RFI for PA does not result to be numerically higher.

Discussion

Line 179-181: The sentence is too long and not clear. You should put a semicolon.

Line 197-199: “while also…………………….bacterial cell”. Not clear. You should rewrite.

Line 201-203: not clear”……..but are all able to inhibit bacterial grow at the concentrations, as exhibited by the MBC values (………..respectively).

Materials and methods

Line 335-338: ..”with increasing concentrations of each of phenolic acids” You repeat this sentence in lane 337. You should put it only once. The 24- well plates……………. “You should indicate the concentrations from and to.

Line 343: ..treatments were bactericidal (>2)? or <?.

After a minor revision I recommend to the Editor the publication of this work.

Author Response

Responses to Reviewer 2 comments (Manuscript ID: antibiotics-945777)

Comments and Suggestions for Authors

The content and the results of the submitted manuscript are interesting, new and contribute to a wide knowledge on the use of polyphenolic compounds as antimicrobial agents against the pathogen Samonella enterica serovar Typhymurium so giving an insight for novel therapeutic systems as promising alternatives to the conventional antibiotics. The paper is well presented and the methods used adequate for describing the experiments. However some points are not clear.

Results

Line 107: “ ……as they fell on or below a ratio of 2”.

Response: We thank the reviewer for pointing out this grammatical error. The word has been corrected. (Line 117)

Line 136-137: “At 8 hr, RFI for both PA and VA decreased, only remaining slightly numerically higher than control”? From Figure 3 RFI for PA does not result to be numerically higher.

Response: We thank the reviewer for pointing out this oversight on our part and apologize for any misunderstanding. The phrase was modified to prevent any future misunderstanding (Line 146).

Discussion

Line 179-181The sentence is too long and not clear. You should put a semicolon.

Response: We appreciate this suggestion. In accordance with it, we have separated the sentence into two separate ones (Line 189).

Line 197-199: “while also…………………….bacterial cell”. Not clear. You should rewrite.

Response: We value the reviewer’s input with regards to this sentence. We have taken their advice and re-worded the entire sentence and the remaining section of this paragraph accordingly for clarification (Line 207), including an update in the citation (Line 572 and 576). 

Line 201-203: not clear”……..but are all able to inhibit bacterial grow at the concentrations, as exhibited by the MBC values (………..respectively).

Response: In response to this reviewer’s comment, we have modified the text for clarification in accordance to the reviewer’s suggestion (Line 217).

Materials and methods

Line 335-338: ..”with increasing concentrations of each of phenolic acids” You repeat this sentence in lane 337. You should put it only once. The 24- well plates……………. “You should indicate the concentrations from and to.

Response: We greatly appreciate the reviewer’s time for pointing this detail out. We have modified the text to eliminate repetition and added the information that they have requested (Line 362 and 363).

Line 343: ..treatments were bactericidal (>2)? or <?.

Response: We thank to reviewer for this comment. In response to this comment, it has been corrected in the text of the manuscript as “less than” (<) (Line 369).

Thank you

Best regards

Round 2

Reviewer 1 Report

No further comments